# Microgreens—A Comprehensive Review of Bioactive Molecules and Health Benefits

**DOI:** 10.3390/molecules28020867

**Published:** 2023-01-15

**Authors:** Maharshi Bhaswant, Dilip Kumar Shanmugam, Taiki Miyazawa, Chizumi Abe, Teruo Miyazawa

**Affiliations:** 1Centre for Nanoscience and Nanotechnology, Sathyabama Institute of Science and Technology, Chennai 600119, India; 2New Industry Creation Hatchery Center (NICHe), Tohoku University, Sendai, Miyagi 980-8579, Japan

**Keywords:** microgreens, bioactive compounds, cotyledons, metabolic diseases

## Abstract

Microgreens, a hypothesized term used for the emerging food product that is developed from various commercial food crops, such as vegetables, grains, and herbs, consist of developed cotyledons along with partially expanded true leaves. These immature plants are harvested between 7–21 days (depending on variety). They are treasured for their densely packed nutrients, concentrated flavors, immaculate and tender texture as well as for their vibrant colors. In recent years, microgreens are on demand from high-end restaurant chefs and nutritional researchers due to their potent flavors, appealing sensory qualities, functionality, abundance in vitamins, minerals, and other bioactive compounds, such as ascorbic acid, tocopherol, carotenoids, folate, tocotrienols, phylloquinones, anthocyanins, glucosinolates, etc. These qualities attracted research attention for use in the field of human health and nutrition. Increasing public concern regarding health has prompted humans to turn to microgreens which show potential in the prevention of malnutrition, inflammation, and other chronic ailments. This article focuses on the applications of microgreens in the prevention of the non-communicable diseases that prevails in the current generation, which emerged due to sedentary lifestyles, thus laying a theoretical foundation for the people creating awareness to switch to the recently introduced category of vegetable and providing great value for the development of health-promoting diets with microgreens.

## 1. Introduction

Globally, in the last decade, especially during- and post-COVID-19 pandemic, the growing interest of society in eating fresh, healthy, and functional foods, such as sprouted seeds and microgreens, has been on the rise [1]. In addition to providing valuable nutritional compositions, they also meet consumers’ preferences for novelty and palatability. Moreover, they are a very attractive product for producers as they require minimal production requirements and reach their maximum consumption within a relatively short time frame [2]. Microgreens, known as “vegetable confetti”, are developed from various commercial food crops, such as vegetables, grains, and herbs that consist of fully developed cotyledons with or without the partially expanded true leaves [3]. The exact portion of the emerging stem of the tender plant, along with the cotyledon leaves and the probable true leaves, are being harvested 7–21 days after germination [4]. These functional micro-vegetables are usually 2–8 cm in height and have intense sensory attributes, such as flavor, texture, aroma, appearance, and exotic colors, irrespective of their small expanse. They are also overloaded with an abundant level of various phytonutrients, varying according to the nature of the plants that are selected to produce the microgreens [5] Due to its attributable presence of various health-promoting phytonutrients, such as antioxidants, vitamins, minerals, phenolic compounds, and much more health-promoting compounds, they are considered the next generation of “superfoods” or “functional foods”.

Microgreens have piqued consumer interest, especially chefs’ of high-end restaurants who use various microgreens, primarily as garnishing elements to enhance salads, soups, sandwiches, and other culinary inventories. However, due to their interesting quality traits, their use has been extended to enrich the diet of a particular group of demanding consumers [6]. They are also preferred as a source of raw foods by various vegans who are specific in consuming nutrient-enriched dietary food. Additionally, they are distinctively peculiar in their growth pattern where they do not require much land space for their cultivation but can be produced in a cramped-up, little space and, thus, can be adapted by any individual without a professional maintenance [7].

Researchers have been exploring their diverse chemical compositions, as well as their respective functions and importance in human health due to their increased demand in using them for a variety of culinary purposes among consumers, urban farmers, greenhouse growers, and grocery stores [1]. It is a well-known fact that a huge expanse of our population is suffering from malnutrition, where both adult and young people are affected due to a lack of micro and macronutrients [8,9]. An adequate amount of the different nutrients is to be provided where it is considered an essential task to drive out the malnutrition known as the hidden hunger [10]. It is known that eating sufficient fruits and vegetables as part of a healthy diet reduces the risk of many chronic diseases, including cardiovascular disease, type 2 diabetes, some cancers, and obesity. However, only a small portion of adults fulfill the recommended daily amounts of fruits and vegetables [11]. Likewise, the problem of consuming a high number of fruits and vegetables is posed a serious threat to human health instead of coping with the protective benefits against some chronic diseases. This is due to the increased consumption and reliability of the chemical fertilizers and pesticides used for their growing, posing a serious threat to the entire food chain in a slow and steady manner [12,13,14]. The observed toxicity over the prolonged consumption of vegetables and fruits that have been cultivated according to modern agricultural practices has diverted the attention of people towards the safe, biological growth techniques and harvesting procedures of microgreens.

Owing to these complications in the agricultural and health sector, microgreens are attributed as functional foods which are rich in various phytochemicals and nutrients, which are likely to be expected to provide a certain solution to the occurrence of some chronic diseases, such as malnutrition, cardiovascular diseases, obesity, diabetes, cancer, neurodegenerative disorders, etc. This multifaceted but one-stop solution of consuming microgreens has attracted health professionals, nutritionists, researchers, and public persons to evaluate their efficiency in treating such disorders. This review mainly focuses on the chemical composition of microgreens and the use of microgreens in health-promoting applications. We have also introduced a small section on the growth conditions of microgreens which have been heavily discussed in other reviews.

## 2. Literature Search

The following search algorithm: (microgreens AND (“sprouts” OR “baby greens”)) was used in four databases (PubMed, Web of Science Core Collection, Scopus, and Google Scholar) for identifying relevant articles. Additionally, a supplementary literature search was done, examining the reference lists of all relevant studies and pertinent review articles to identify articles not identified in our electronic search.

## 3. Sprouts vs. Microgreens vs. Baby Greens vs. Mature Plants

In recent years, humans started giving more importance to their health, especially due to the outbreak of various kinds of disorders and constant deterioration in their health [15]. Consequently, they ate fresh fruits and vegetables and other kinds of green leafy vegetables as part of their daily diet to ensure adequate nutrition. This phenomenon has led to the search for more functional foods that provide an abundance of nutrients economically and without any or with little interference from modern agrochemicals. The term “organic” has become more popular due to the increasing difficulties associated with modern agricultural practices; traditional foods are being brought back to life in order to produce safe and healthy foods. In this light, the old practice of consuming green leafy vegetables has given rise to new yet old food products called baby greens, sprouts, and microgreens. These three micro vegetables are believed to act as functional foods that provide exceptional nutritional elements, thus, imparting health benefits to the consumers.

Interestingly, sprouts and microgreens are so advantageous that their growth period is much shorter, and their maintenance is considerably lower compared to matured green plants and their produce, such as vegetables and fruits [16]. Sprouts are the easiest to cultivate when grown in the dark without a need for a growing system, such as soil and other nutrients, agrochemicals, etc. [17]. The entire plant that has been grown is consumed, which consists of the first-ever seedling along with the radicle portion. They are generally eaten raw and contain a lot of dietary fiber along with the abundance of stored plant phytochemicals that play a major role in improving human health [17]. As for microgreens, they are considered superfoods that can be produced in urban and peri-urban settings in a limited space with a short growth cycle and a minimum to no use of external nutrients for growing [18]. They are grown in the presence of various kinds of light and growing mediums resulting in fully-developed cotyledons with one or two true leaves. They are believed to provide and supply enough nutrients; these are now studied by various researchers considering whether their sole consumption can replace the intake of our usual food regime of vegetables and fruits. On the other hand, baby greens and mature plants are somewhat similar to each other; they both require proper light, growth medium, an expanded time period, and external nutrients to grow [19]. Additionally, baby greens differ from their mature counterparts because they can be consumed raw, whereas matured plants usually require cooking before consumption [19]. Furthermore, baby greens are young plants with tender, true leaves when compared to mature plants with well-defined root and shoot systems. In common, microgreens, sprouts, and baby greens are the new facets for health specialists, requiring mild treatments for their production and used as raw leafy vegetables. The different characteristics of sprouts, microgreens, baby greens, and mature plants are discussed in Table 1.

## 4. Different Varieties of Microgreens

Microgreens (Figure 1) are produced both at a small-scale level and along with large-scale production of commercial vegetables and edible flowers. Upon their meteoric rise and demand, various varieties of commonly grown vegetables were used to cultivate microgreens that belong to various families, such as Amaranthaceae (amaranth, beet, quinoa, spinach, buckwheat, chard), Amaryllidaceae (garlic, onion, leek), Apiaceae (parsley, carrot, fennel, celery, dill, carrot, chervil, cilantro, coriander), Asteraceae (lettuce, radicchio, chicory, endive, tarragon, common dandelion), Boraginaceae (phacelia), Brassicaceae (radish, watercress, arugula, broccoli, cauliflower, cabbage, chicory, wild-rocket), Convolvulaceae (water convolvulus), Cucurbitaceae (melon, cucumber, squash), Malvaceae (jute mallow/Nalta jute), Poaceae (corn, lemongrass), Lamiaceae (chia), Leguminosae (chickpea, alfalfa, bean, green bean, fenugreek, fava bean, lentil, pea, clover), Onagraceae (evening primrose), Portulacaceae (common purslane, moss-ross purslane) [1,2,19,23]. The interest in different promising genotypes of microgreens offers different appearances, flavors, textures, phytochemical compositions, and nutritional values. Microgreens vary in taste. For example, some can be bitter, spicy, mild, bland, or even sour tasting [24]. Other herbaceous species commonly used to produce microgreens are cereals (oat, soft wheat, durum wheat, corn, barley, rice), Oleaginous plants (sunflower), and even fiber plants, such as flax, as well as many aromatic species such as basil, chives, cilantro, and cumin. The microgreens species are selected from an agronomic and commercial point of view and are strongly characterized by the availability of good quality seeds, homogeneous nature, germination capacity, hygienically safe, and, at the same time, are available at a low cost. Above all, the selected species should be available around the year with no particular thermal and environmental needs during the germination phase. The species selection is also sometimes based on shape, color (green, yellow, purple, red, crimson, and multicolor), texture (juicy, crunchy), and shelf-life [25].

## 5. Nutrient and Phytochemical Composition of Microgreens

As an emerging food source, the chemical composition of microgreens is yet to be explored, and very little information is being documented. It is said that microgreens are largely associated with micro and macronutrients, such as Fe, Zn, K, Ca, N, P, S, Mn, Se, Mo, and other. Apart from these mineral components, microgreens are rich in biological phytochemicals, which have an immense potential to enhance human health and also aid in improving diseases. The major bioactive compounds, such as ascorbic acid, phylloquinones, α-tocopherol, β-carotene, phenolic antioxidants, carotenoids, anthocyanins, glucosinolates, and sugar content, are reported to be present in the microgreens in larger contents. A comparison of red cabbage’s (*Brassica oleracea* L. var. capitata) phytochemical concentrations during the microgreen and adult growth stages showed that the microgreen stage has a high amount of phylloquinone (2.8 μg/100 g FW), β-carotene (11.5 mg/100 g FW), and glucoraphanin (4.8 μmol/g DW) compared with matured stage [26]. However, anthocyanins are in higher quantity at the mature stage compared to the microgreen stage [26]. The various bioactive phytochemicals (Table 2) present in the various types of microgreens are discussed in detail in the following subsections:

### 5.1. Ascorbic Acid—Vitamin C

Ascorbic acid is an essential bioactive phytochemical, also known as vitamin C, and is essential for the body’s functioning. It is also categorized as an antioxidant that helps in various metabolisms of humans. Di Bella et al. 2020 investigated the ascorbic acid content in the microgreens and noticed changes in the levels of ascorbic acid at various stages of plant growth, suggesting that the ascorbic acid level was potentially higher in the microgreen stage of the plant development than in the other stages, such as sprouts, baby greens, and mature plants [19]. This was later investigated in the application of nutritional stress in the microgreen production and its impact on the ascorbic acid level and phenolic acid content, suggesting that the total ascorbic acid content has been positively increased by 187% upon the application of nutritional stress [38].

### 5.2. Phylloquinone—Vitamin K

Phylloquinone is a direct and circulating form of vitamin K, which is mainly present in vegetables, fruits, and other green leafy vegetables. Xiao et al., in 2015, evaluated the sensory attributes and the various chemical components present in six different species of microgreens, namely Dijon mustard (*Brassica juncea* L. Czern.), opal basil (*Ocimum basilicum* L.), bull’s blood beet (*Beta vulgaris* L.), red amaranth (*Amaranthus tricolor* L.), peppercress (*Lepidium bonariense* L.), and China rose radish (*Raphanus sativus* L.). Their results showed that the phylloquinone content did not vary from one species to another and was found to be within the range of 2.1 and 4 g/kg of the microgreen produce [39].

### 5.3. α-Tocopherol—Vitamin E

An α-tocopherol is an extremely important phytochemical that is present in microgreens. They are involved in many of the body’s functions, especially in nerve impulses, muscle movements, boosting the immune system, limiting free radical formation, and many more important activities [40,41]. Various researchers reported that microgreens contain a substantial amount of vitamin E, thus, helping consumers to improve body functioning. For instance, the nutritional content was evaluated in a group of six genotypes of microgreens that belong to three species and two different families, and it was found that the α-tocopherol content was substantially higher than the other reported microgreens [34]. They reported that the six microgreens had a wide range of concentrations of tocopherol (11–76 μg/mg of microgreen).

### 5.4. β-Carotene—Pro Vitamin A

A β-carotene is a red–orange organic compound that acts as the precursor of vitamin A and is a plant metabolite that is especially present in red-, yellow-, and orange-colored plants. They play a major role in the inhibition of free radicals, induction of apoptosis in cancer cells, and the enhancement of natural killer cell production, thus improving the immune system [42,43]. These are present in the microgreens and are a great source of pre-vitamin A content for consumers. The phytochemicals present in 10 different culinary microgreens obtained using HPLC-DAD showed that the selected microgreens were excellent sources of β-carotene and other phytocompounds, such as vitamin E and ascorbic acid, revealed that β-carotene concentration range between 3.1–9.1 mg/100 mg of the microgreen and a maximum of the β-carotene content was found in the fennel, radish, and mustard [32].

### 5.5. Phenolic Antioxidants and Sugar Content

Phenolic antioxidants are secondary metabolites that are present in the microgreens that help in promoting metabolic activity, preventing free radical oxidation, and reducing inflammation [44]. The antioxidant properties and organoleptic properties of different kinds of microgreens that are procured from local and commercial farms are investigated for phenolic antioxidants, such as tannins, phenolic acids, anthocyanins, and other antioxidants associated with organoleptic activity, such as flavor, taste, and color of the microgreens [23]. It was also found that the total phenolic content in the microgreens ranged between 10.71–11.88 mg/g, especially in broccoli, which was 10 times higher than that of the respective mature counterparts and sprouts. These phenolic contents in the microgreens are responsible for the improvement of glucose homeostasis and various other metabolic reactions in the body [23]. Gao et al., in 2021, investigated the production of high-quality broccoli microgreens and the variable effects of the low-light intensity on the phytochemical content of the plant produce [45]. Their study suggested that the light intensity plays a major role in the soluble sugar content and found that the free sugar content was 5.44 mg/g of the microgreen grown under 50 µmol/m^2^/s, while the lowest content found among the variable irradiance was 70 and 90 µmol/m^2^/s.

### 5.6. Anthocyanins and Glucosinolates

Anthocyanins are a group of organic compounds present in plants and their subsequent parts, such as leaves, fruits, and vegetables, that impart blue, purple, and red pigments and are responsible for various activities, such as antioxidant, anti-inflammatory, anti-cancer, and anti-viral properties [46]. These flavonoid compounds are seen in several types of microgreens that improve various metabolic conditions present in the human body [47]. The bioactive profile of three different Brassica L microgreen species that are grown in peat-based media was investigated for anthocyanins, accumulated in cell vacuoles of the red cabbage and rocket microgreens and showed 11.9% and 20.2%, respectively [47]. Glucosinolates are also secondary metabolites that are synthesized by microgreens and are attributed for their health benefits, such as antioxidant capacity, anti-inflammatory capacity, and other [48,49]. The bioactive properties of glucosinolates present in four different Brassicaceae microgreens and their metabolomics were studied using UHPLC-QTOF mass spectroscopy, a successful technique for the fast and high-resolution separation with high sensitivity, which showed that there are around 22 different kinds of the glucosinolates present, among which the most represented class of compounds are glucoraphanin, glucobrassicin, gluconapin, and 4-hydroxygluccobrassicin [48,49]. These studies suggest that red cabbage contains the highest amount of these compounds (with 197.8 mg/100 g dry weight of the microgreen) compared with the other four microgreens used.

### 5.7. Micro and Macroelements

Micro and macroelements are essential to impart a healthy lifestyle which aids in the metabolic processes, energy production, and combatting COVID-19 [50,51,52]. Microgreens are seen as a major source of various important micro and macronutrients with a significant difference in the overall composition of the nutrient content among various microgreen types. Renna and Paradiso, in 2020, investigated the nutritional content of three different microgreens, namely cauliflower, broccoli, and broccoli rabe, belonging to the Brassica family and cultivated using three different molar ratios of NH_4_:NO_3_ nutrient solutions [53]. It is evident from this study that these three microgreen species are rich in mineral elements, such as Na, Cu, Mn, Ca, Mg, K, Zn, and Fe. Apart from these minerals, they are also rich in the macroelements, such as proteins, dietary elements, α-tocopherol, β-carotene, and others. Similarly, a group of six genotypes of microgreens that belongs to three species and two different families is characterized by mineral nutrient profile (*Cichorium intybus* L, Molfetta, cultivar ‘Italico a costa rosa’ lettuce from Asteraceae and broccoli from Brassicaceae). These were compared with their individual corresponding mature plants, and it was found that the mineral contents were much higher in the former vegetable groups than in the latter vegetables [34]. Besides, they found from the nutrient profile of each microgreen that they are abundant in the nutrients, such as K, P, Ca, Zn, Fe, Mg, and Na. This suggests that in the future, microgreens can be administered to people who are deficient in such nutrients rather than administering their chemical form. Furthermore, microgreens can be customized and produced according to consumer needs.

## 6. Growth Conditions for the Production of Microgreens

Microgreen production is an increasingly popular phenomenon due to the microgreens’ vivid color, intense flavor, and crunchy texture. They should be consumed regularly due to their nutraceutical properties promoting immense health benefits to the consumers [26,54]. Briefly, the requirements that are considered for growing microgreens are discussed in Table 3. Their high demand in the pharmaceutical and food industries is due to the remarkable content of bioactive compounds. Their production on both a commercial and small-scale basis has been gradually increasing in urban areas [55]. Microgreens can be produced easily due to their short span of growth, simple growing techniques with soil or soil-less system, artificial light techniques, etc. [55]. In this section, we briefly discussed various important factors that play a major role in the production of high-quality microgreens.

### 6.1. Seed Treatment

Microgreens are commercially produced either in a soil-based cultivation system or in a soil-less cultivation system with an alternative variety of substrates maintained in a greenhouse with a proper irrigation system. The efficient growth of microgreens was majorly dependent on the germination of seeds and also characterized by the seed quality. Seed density and treatment play an important role in the growth of healthy microgreens and vary from one species to another [61]. Lee et al., in 2004, investigated different seed treatments in the production of beet and chard microgreens [56]. They tested different seed treatments, such as seed priming (seed soaking in sodium hypochlorite, water, hydrochloric acid, and hydrogen peroxide) and matric priming (seed germination in fine vermiculite at 12 °C and –1 MPa for six days), and found that germination percentage was greatly affected by the matric priming, rather than different soaking methods, and showed a 0.33–2.79 fold increase in shoot weight of the microgreen [56]. Hoang and Thuóng, in 2020, investigated the seed density and substrates to achieve a larger yield of radish microgreens in controlled greenhouse conditions. The study resulted in a significant increase in the fresh yield of the microgreens when seeds were sown in a ratio of eight seeds per cell (109 g of radish seeds) [62]. It is evident from these studies that seed density and quality play an important role in the attainment of higher yields of microgreen produce, and these parameters change from one species to another.

### 6.2. Light

Light is a substantial parameter that greatly affects microgreens production. Light is directly involved in the microgreen yield and nutritional composition, and its variations may affect the quality of microgreens at a greater level [63]. Brazaityte et al., in 2015, investigated the effect of solid-state LED light over carotenoid content in Brassicaceae species microgreens [64]. They evaluated the spectral wavelength and the irradiance level of LED light in the growth of red pak choi and tatsoi microgreen species. They found that the lowest irradiance level (110 μmol/m^2^/s) accumulated a lower amount of carotenoids, and the highest irradiance level (545 μmol/m^2^/s) also accumulated a lower amount of carotenoids in mustard and red pak choi by an approximate level of 24% [64]. Likewise, the spectral component of green color irradiation for the microgreens growth has a positive note in carotene content, whereas the orange spectral light had a negative effect on the carotene content of the microgreens [63,64]. Similarly, the effects of different dosages of blue light and their effects on the concentration of phytochemical compounds, such as carotenes, violaxanthin, zeaxanthin, lutein, chlorophyll, carotenoids, and tocopherols, were also evaluated [65]. All these compounds were accumulated in about 1.2–4.3 times higher quantity except for the tocopherol compound, which is independent of the light reactions [5,33,65], suggesting that metabolite concentrations were significantly altered irrespective of their mechanisms being light-dependent or not.

### 6.3. Growth Medium

An interest in microgreens production has increased 1.2–4.3 times recently due to the important fact that people tend to pay more and more attention to organic foods consumed daily [7]. On that basis, conditions at which microgreens are produced, especially the growth medium, were also being changed from the traditional soil-based production. Instead of soil use, other growth mediums containing coconut fiber, perlite, vermiculite, and peat moss are used for the successful production of the microgreens [66,67]. In that light, another important growth medium is the hydroponic system, where microgreens are grown immersed in a combination of water and other essential nutrient solutions. Muchjajib et al., in 2015, investigated the effect of various organic Thailand-based alternative growth mediums or substrates instead of peat for the successful cultivation of various microgreens [68]. There are several studies using local biomaterials as growth mediums, such as sand, coconut coir dust, peat, vermicompost, sugarcane, filter cake, and their different combinations, which were examined for the maximum yield of the microgreens [61,69]. It is found that vine spinach microgreen grown under the coconut coir dust yielded a maximum yield of 5.17 kg/m2 [68]. It is also evaluated that the nutritional composition of the microgreens that are grown in different growth media are affected by the microbial attacks of *Escherichia coli*, yeast, *Salmonella*, mold, and *Staphylococcus aureus*, which were at a permissible and safe limit [70,71]. Similarly, comparing the efficiency of the nutritional composition of microgreens grown in two equally important growth media, namely soil and hydroponic (water) systems, concluded their elemental and nutrient analysis that the water system is much more efficient than traditional soil-based growth media [72].

### 6.4. Microbial Colonization and Pests and Diseases

Crop failure due to microbial and pest attacks is a major problem in the agricultural sector that can topple down the produce and, thus, cause a huge loss to farmers and agriculturalists. However, in the case of microgreens, since production time is shorter by about 7–21 days, there are no major pest attacks that entirely affect microgreen production. However, there is substantial microbial contamination that occurs, especially due to the careless practices of sowing denser seeds in the trays or the channels with prevailing high humid conditions, which are observed especially in the hydroponic systems [70,71]. Studying colonization dynamics of the *Escherichia coli* O157:H7 inoculation and quantified microbial colonies that invaded the plant (microgreen) system revealed that plant tissue was attacked externally over the surface of the shoots, and also the internalization via stomatal pores was observed in nine different microgreen varieties that were investigated [73]. It is hypothesized that the possible reasons for the invasion of microbial populations into microgreens might be due to imperceptible contamination occurring in the irrigation system, inoculum dosage, higher seed density, use of contaminated seeds, high humidity rate, and other external environmental aseptic conditions [73]. This microbial colonization not only affects the production of microgreens but also affects the nutrient composition and, finally, may cause food-borne diseases upon the consumption of microgreens by the consumers [7]. Microgreens, especially such varieties as Brassicaceae, which have a high seed density per tray or channel, have been vulnerable to pythium root rot which is caused by fungal species, such as *Pythium aphanidermatum* and *Pythium dissotocum*. It has been investigated that bio fungicides and their probable effects on the microgreens, which are invaded by the fungal species, especially in cases of pythium root rot and dampening of microgreen greenhouses. This condition can be avoided by the addition of bio fungicides into the irrigation system rather than synthetic fungicides, which had a negative impact on the yield of the microgreens [74]. Similarly, microgreens produced by the inoculation of the bio fungicides resulted in a 59% increase in the biomass produced.

### 6.5. Harvesting and Post-Harvesting Techniques

Microgreens contain cotyledons and hypocotyls when they are as young as 7–14-day seedlings harvested at the right time, which usually differs from species to species. These young shoots are needed to be packaged due to their high marketability. Various types of microgreens have been commercially produced and also been produced and harvested at a household level [75]. Hence, the usual harvesting techniques, the hand-picking method, and, in the case of large-scale commercial greenhouses, both hand harvesting and mechanical harvesting, are being practiced. Harvesting is a simple process in the cultivation of microgreens during which the seedlings are collected from the growth medium or are packaged together with the plated growth medium after the seedlings reach a certain desirable height [75]. These harvested crops are then washed and get subjected to various good handling practices for post-harvesting techniques in order to maintain the food safety grades. The harvested plants are then packed in polythene bags or clamshells, which are pre-treated to avoid any cross-contamination.

Due to the perishable characteristics of the green and fresh microgreen produce, it is very important for the producer to improve the marketability rate by subjecting it to various post-harvesting techniques so that the shelf life of the microgreens is increased to an extent by maintaining the respiration rates and avoiding any possible contamination [76]. Generally, microgreens are subjected to washing, drying, freezing, microwaving, and chemical treatments in order to maintain their phytochemical content and overall nutritional value even after a few days after harvesting, i.e., till they reach the hands of the customer [71]. Extending the shelf life of microgreens is an important aspect that is being studied by various groups of researchers, and two important parameters, namely the atmospheric condition of packed microgreens and the storage temperature, play a major role in it. These two above-said parameters were investigated, and it was found that as time progresses, the shelf-life and quality of microgreens are greatly affected even when they are stored at a temperature of 4 °C [77]. The microgreen species stored at a relatively higher temperature of 10 °C have a much lower shelf-life extendibility [77]. Moreover, a high respiration rate affects the visual quality of microgreens, which might directly have a negative impact on customers’ perceptions. The exposure of microgreen produce to the light in supermarkets and vegetable stores plays a detrimental role in the nutritional quality of the microgreens; hence, the microgreens should be stored and maintained very carefully until consumption. The decrease in the oxygen rate and increase in the CO_2_ rate also help extend the shelf-life of the microgreens. Apart from these methods, various techniques, such as chlorine washing, ascorbic acid, citric acid, and ethanol, are applied in order to prevent the physiochemical properties of the microgreens during their storage [77].

## 7. Effect of Microgreens on Metabolic Health-Promoting Applications

Foods have indeed been essential in the development of human culture [78]. Foods contain calories and vital nutrients essential for human growth, development, and survival [79]. In addition to providing nourishment, food also helped people in many cultures avoid and treat numerous health issues [80]. The modern era of food science and nutrition reflects the growth of humanity, and advancements have been made thanks to the infusion of knowledge from such fields as medicine, biology, and biochemistry [81]. The shifts in this paradigm are the end consequence of numerous years of science-based work. For instance, findings from the community and experimental studies typically suggest the health-protective benefits of diets high in foods originating from plants [82]. The emphasis on nutrition and diet also focuses on preventing micro-nutrient deficiencies (vitamins, minerals, etc.) and mitigating the effects of chronic diseases, such as obesity [83].

Chronic metabolic disorders, which affect human health over a longer period, have always been a looming issue in the health sector. Metabolic disorders usually do not pose an immediate threat to human health but cause other health issues over time and prevail for longer, thus increasing the risk factors among people [84]. Metabolic disorders, such as obesity, cardiovascular diseases, neurodegenerative disorders, diabetes, and co-morbidities, have become increasingly common among people due to aggressive lifestyle changes [85,86]. Microgreens, which are tiny harvested vegetables with a high density of nutrients, minerals, and phytochemicals, are now in the limelight and are used for various culinary enhancements. In this section, we discussed the application of microgreens in improving some chronic metabolic disorders and their efficacies in reducing disease, as depicted in Figure 2.

### 7.1. Diabetes

Diabetes is a chronic metabolic syndrome, which is characterized by the cells becoming resistant to insulin or by the inability of the pancreas to secrete sufficient insulin levels. According to International Diabetes Federation (IFD), in the next 20 years, the number of individuals (20–79 years) who have diabetes is expected to increase to 783 million from the current 573 million [87]. Diabetes is characterized by a high blood glucose level and is usually treated or maintained by lowering blood glucose levels through vigorous diet regimes, insulin injections, supplementing insulin secretion, and increasing insulin sensitivity [88,89]. However, prolonged pharmacological intake may pose adverse effects; thus, the intake of vegetables and fruits with no or low sugar content is usually prescribed [90]. The increasingly popular microgreens are highly nutritious and are suggested to have more efficacy and high potential to reduce diabetes [91]. The fenugreek microgreen extract (2 mg/mL) inhibited α-amylase by 70% in HepG2 cells and enhanced glucose uptake in L6 cells by 44% in the presence of insulin [91]. In vitro regulation of anti-diabetic activity is probably due to the high levels of phenolic content, flavonoids, and antioxidants in the fenugreek microgreens. This extract also inhibited the non-enzymatic glycation of protein. Furthermore, lyophilized broccoli microgreen powder 2 g/kg bodyweight has shown a hypoglycaemic effect in mice fed with a high-fat diet and/or with streptozotocin [92]. Further, barley (*Hordeum vulgare*) microgreen, a member of the grass family, which contains phytochemicals, such as 3′-Benzyloxy-5,6,7,4′-tetramethoxyflavone, 5β,7βH,10α-Eudesm-11-en-1α-ol, 4′,6-Dimethoxyisoflavone-7-O-β-D-glucopyranoside, citronellyl tiglate, 3,4-dihydrocoumarin, and phytanic acid, improved glucose metabolism in streptozotocin and/or aflatoxin induced diabetes condition in male albino rats [93,94]. Additionally, sperm count significantly decreased in aflatoxin and/or streptozotocin groups but significantly increased in groups treated with barley microgreens. Barley microgreens-treated groups also exhibited a significant reduction in sperm morphological defects and chromosomal aberrations in comparison to untreated animals [94]. With the limitation of number of strong scientific evidence with microgreens in the reduction of diabetes or maintaining glucose homeostasis, the presence of high phytochemicals in microgreens suggests that they need to be further explored to understand their effect and efficacy in regulating body carbohydrate metabolism.

### 7.2. Chronic Kidney Disorder

A kidney disorder is an emerging global issue that is categorized under chronic metabolic syndrome due to its relation to other lifestyle-related diseases [95]. Patients with chronic kidney disorders are restricted in the intake of high-potassium-content foods. Therefore, they are advised to consume vegetables and fruits with less or no potassium content. Kidney-impaired patients are at high risk if they consume potassium foods, but one needs to be fed with an adequate amount of potassium in order to maintain normal or lower blood pressure. These patients are also sensitive to heart-related disorders, hypertension, diabetes, hyperlipidemia, and various other sedentary lifestyle diseases [95]. The fact that potassium is highly present in vegetables, attention has been diverted to microgreen production, whose nutritive value can be altered by changing the cultivation parameters, thus, producing low-potassium-content microgreens. Renna et al., in 2018, produced a novel chicory and lettuce microgreen in a hydroponic system with substantially lower potassium content [96]. This low potassium content was achieved by the absence of potassium salts in the nutrient solution during the growth of the microgreen production. They concluded that the production of microgreens with the absence of potassium in the nutrient solution had no effect on the nutritional quality, appearance, texture, and color but had a slight reduction in the biomass yield of the microgreens. Nevertheless, the produced microgreens had a permissible amount of potassium content, which is recommended by the doctors. Intake of microgreens with lower potassium content can aid patients with impaired kidney functions who were accustomed to taking vegetable-based foods.

### 7.3. Cancer

The second-leading cause of death after heart disease is cancer, a serious issue for public health worldwide [97]. Although the prevention mechanism against cancer remains unclear, with an increase in the consumption of a diet rich in fruits and vegetables, especially in cruciferous plants, their bioactive compounds have been postulated for their protective effects [26,98,99]. Microgreens are recommended as a functional food that is highly rich in bioactive compounds, such as carotenoids, chlorophylls, tocopherols, glucosinolates, polyphenols, and ascorbic acids [100]. Brassicaceae microgreens are traditionally recommended for cancer patients, obese, and chronic heart disease patients [26]. The major types of cancer that prevail in various people are colon, lung, breast, gall bladder, and liver cancers. These cancer patients are recommended a highly nutritious and regular consumption of vegetables [101,102]. Recently, the relationship between diet regime and cancer has been studied widely, and the intake of cruciferous vegetables is highly recommended for the prevention of cancer [103]. Fuente et al., in 2020, investigated the antiproliferative effect of four Brassicaceae microgreens (radish, broccoli, kale, and mustard), which were produced by the hydroponic systems and were evaluated in the colon Caco-2 cells vs. normal colon CCD18-Co cells [104]. The cells were treated with bio-accessible fractions of four microgreens for 24 h, and their bioactivity was compared with the colon cancer chemotherapeutic drug 5-fluorouracil. The cells treated with microgreens showed an increased reactive oxygen species, decreased glutathione intracellular content, general cell cycle arrest in G2/M, and apoptotic cell death [103,104]. Furthermore, Thai red-tailed radish microgreen extract from cold-plasma-treated seeds showed an anti-migratory effect in human breast adenocarcinoma (MCF-7) and human hepatocellular carcinoma (HepG2) cells and induced apoptosis and prevented cancer cell proliferation through upregulation of *Bax* and *Caspase-3* expression [105]. Additionally, MMP-2 and MMP-9 genes associated with tumor invasion and metastasis are significantly inhibited by the red-tailed radish microgreen extract [105]. Phospholipid bilayer vesicles, known as extracellular vesicles (EVs), are released by cells and contain organic cargo. The study was conducted using Raphanus sativus L. var. caudatus alef microgreens as the source of EVs, and Tai rat-tailed radish microgreens with their nano-vesicular shape were made up primarily of natural macromolecules, specifically a protein with a secondary structure of a pleated sheet [106]. Additionally, EVs outperformed microgreen extract in terms of safety and increased selectivity when used against HCT116 colon cancer cells [106]. These microgreens can be incorporated into the balanced diet regime of cancer patients in order to reduce the impact of chronic degenerative diseases, such as cancer.

### 7.4. Cardiovascular Diseases

Cardiovascular diseases account for one of the major causes of all deaths among other chronic diseases [107,108]. Cardiovascular disease is a chronic metabolic disorder that is caused by hypercholesterolemia [109]. Various epidemiological studies revealed that an increase in the intake of fruits and vegetables lowered the occurrence of cardiovascular diseases [110,111]. As a part of the strategy to promote health and prevent deadly cardiovascular disease, the consumption of microgreens is highly recommended due to their high nutrient and phytocompound content. Huang et al., in 2016, investigated the efficacy of microgreens in reducing lipid and cholesterol levels in an in vivo study [112]. They used a high-fat diet-induced obesity model and supplemented it with microgreens along with their diet. In addition to reducing body weight, animals showed a significant reduction in the low-density lipoprotein levels, hepatic cholesterol ester, and expression of inflammatory cytokines in the liver and triacylglycerol levels [112]. In addition, the reduction of inflammatory cytokines in the liver and hepatic cholesterol correlated with the inhibition of the enzymes responsible for triglyceride synthesis, including glycerol-3-phosphate acyltransferase, acetyl-CoA acetyltransferase-3, and lecithin-cholesterol acyltransferase [113] were observed. These results suggest that microgreens can be stipulated for modulating weight control and cholesterol metabolism and ultimately preventing chronic cardiovascular diseases. The therapeutic components in red cabbage or other microgreens need to be further clarified because their composition was distinct from that of their mature equivalents [112].

### 7.5. Inflammation

Inflammation plays a severe role in the pathogenesis of various chronic and acute disorders [114]. Fabaceae microgreens of the species represented promising new sources of ingredients for the fortification of staple foods with bioactive compounds. Traditional medicinal licorice (*Glycyrrhiza glabra* L.) is obtained from the roots of *Glycyrrhiza glabra* L., *Glycyrrhiza uralensis* Fischer, *Glycyrrhiza inflata* Batalin (Fabaceae). The pharmaceutical importance of licorice lies in the great variety of secondary metabolites extracted from roots, with widely reported antimicrobial, antiviral, antitumor, antidiabetic, anti-inflammatory, immunoregulatory, hepatoprotective, and neuroprotective activities [115]. Marrotti et al., in 2020, investigated the effect of licorice microgreen by preparing a root, stem, and leaf polyphenol extract and evaluated the cell proliferation and viability of Caco-2 cells after pro-inflammatory induction of lipopolysaccharides [115]. These results suggest that root extracts of the licorice microgreen contain high anti-inflammatory polyphenols, thus, consequently inhibiting the pro-inflammatory cascade and cytotoxic effects [115]. On the other hand, lyophilized microgreen powder significantly reduced inflammatory markers, such as TNF-α, IL-6, and IL-10. Additionally, short-chain fatty acids and microbial composition of mouse feces also improved after treatment with broccoli microgreens [92]. Furthermore, streptozotocin and aflatoxin increased lactate dehydrogenase activity, and malondialdehyde and oxidative stress in male albino rats was significantly decreased with barley microgreen treatment [93,94]. The impact of red cabbage microgreens powder in reducing obesity-induced hypercholesterolemia in rats fed a high-fat diet had significantly lowered plasma levels of low-density lipoprotein and hepatic triglycerides compared with the rats given a high-fat diet along with mature red cabbage powder. Additionally, mice fed with microgreens displayed considerably decreased gene expression of sterol O-acyltransferase 1 and diacylglycerol O-acyltransferase 1 compared to control groups [112]. Furthermore, inhibition of synthesis of cholesterol esters and triglycerides was caused by the downregulation of the genes for sterol O-acyltransferase 1 and diacylglycerol O-acyltransferase 1 and demonstrated a beneficial effect of red cabbage microgreens on controlling plasma and liver lipid metabolism. Additionally, ingestion of red cabbage microgreens reduced liver mRNA expression of TNF- α and C-reactive protein, indicating an inhibitory impact of red cabbage microgreens on inflammation brought on by a high-fat diet [112].

### 7.6. Obesity

The study led by Li et al., 2021, investigated the preventive effect of obesity using broccoli microgreen juice in C57BL/6J mice [116]. Broccoli microgreen juice was administered by gavage to the obese model mice fed a high-fat diet to develop obesity. As a positive control, melbin was gavaged at 300 mg/kg–bw/d, significantly decreasing the mass of white adipose tissues, body weight, and adipocyte size, along with increased water consumption. Additionally, it lowered insulin levels, lessened insulin resistance, and enhanced glucose tolerance. These findings suggest that gut microbiota short-chain fatty acids lipopolysaccharides inflammatory axis may play a role in the preventive effects of Broccoli microgreens juice on diet-induced obesity. Broccoli microgreens juice can also reduce the accumulation of fat in the liver by increasing the liver’s antioxidant capability. Therefore, based on its opposing effects on obesity induced by the high-fat diet in rats, these data support the consumption of broccoli microgreens juice as a novel functional food for obesity [116].

### 7.7. Iron Deficiency

Iron deficiency is the most common form of nutritional deficiency that affects most of the world’s population, with an impacting factor among vulnerable groups, such as young children, adolescent teens, women of reproductive age, and old aged people. It affects people drastically, as its symptoms are hard to detect, and it has a deleterious effect over time during adulthood. The general treatment for this disorder is a diet abundant in iron-rich fruits and vegetables, along with prescribed iron supplements for the same disorder. The most sustainable way to prevent the disorder is to increase the bioavailability of iron-rich foods in the human body and by stimulating various metabolic processes that enhance the absorption of the iron content from these vegetable sources. Due to the increased culinary trend of microgreens, Khoja et al., in 2020, investigated the bioavailability of the iron content in such varieties of microgreens as rocket, fenugreek, and broccoli and compared them with their respective mature counterparts. They analyzed the mineral content and bioavailability of the microgreens using microwave digestion and ICP-OES (Inductively coupled plasma optical emission spectrometer). It is found from the results that the fenugreek microgreens have higher levels of iron, which establishes a high level of bio-accessibility to the cells. Hence, they concluded that the fenugreek microgreens could be used for increasing iron levels and its enhancer mechanism in the body.

### 7.8. Biofortification of Essential Nutrients

Micronutrients, vitamin C, and antioxidants are crucial for human health. Unfortunately, vitamin C cannot be synthesized by humans and must be obtained from the diet, as severe deficiencies can result in scurvy [117]. However, consumption is frequently erratic, and the vitamin C content of foods varies. Increased micronutrient or mineral concentrations, or biofortification, can enhance the nutritional value of crops and enable more stable dietary levels of these nutrients [117]. The least studied method to boost vitamin C in microgreens is agronomic, particularly when ascorbic acid was added on top of the other two biofortification methods, conventional and transgenic [118]. Biofortification of vitamin C in microgreens of arugula (*Eruca sativa ‘Astro’*) irrigated with four ascorbic acid concentrations and controlled as the supplement concentrations were grown, microgreens’ total vitamin C and ascorbic acid levels increased significantly. In conclusion, biofortification of vitamin C in microgreens through the addition of ascorbic acid is feasible, and ingestion of these bio-fortified microgreens could assist in meeting the daily vitamin C requirements for people, hence lowering the need for additional vitamins [118]. Similarly, the possibility of iron (Fe) enrichment, biofortifying various vegetable microgreens, such as purple kohlrabi, radish, pea, and spinach microgreens, were studied to see how nutrient solutions enriched with iron chelate (1.5 and 3.0 mg/L) affected growth of the plants and mineral concentration [119]. Except for radish, the Fe content in microgreen leaves of the investigated species increased as the quantity of Fe in the medium increased [119]. Particularly, spinach and pea grown in soilless systems are excellent candidates for producing high-quality microgreens with Fe biofortification. Other microelements, such as Mn, Zn, Cu, and other., can also be studied using various microgreens for biofortification.

### 7.9. Other Applications

As microgreens are used more for their nutritional benefits, especially for their organoleptic properties, such as taste, smell, texture, crunchiness, etc., they found application in the culinary industry as an appetizing agent for various food items [26]. Apart from the potential health-promoting effects, very few studies have been conducted to explore the use of microgreens for other purposes that are essential to different people apart from the culinary approaches. The lack of fresh vegetables for people who live in higher altitudes and remote locations, where transport might cause a serious threat to the nutritional benefits, growing microgreens at these locations would provide the balanced nutrition. [120] As example, a variety of microgreens are specially cultivated in the Trans-Himalayan region for the consumption and delivery of highly nutritious sources of food to the Indian military troops. It is a well-known fact that food and nutrient supply to the military personnel stationed at high altitudes is difficult and that their food and nutritional security has been an issue for a long time. Keeping in mind the temperature and other environmental conditions for the growth of microgreens, they produced a variety of microgreens that includes cauliflower, red cabbage, kohlrabi, radish, cabbage, and fenugreek which possess health-promoting phytochemicals and other required micro and macronutrients [120].

Similarly, the production of nutritious microgreens is possible in space [121]. Prolonged research and investigating the adaptability of human life in space through observing astronauts has led to the idea of growing microgreens in space due to their short period of cultivation along with an abundant level of nutrition. It is said that astronauts face a series of health issues due to continuous travel and experiencing different environmental conditions, such as psychological stress, eye health, central nervous system impairment, weight loss due to inadequate supply of nutritious foods, etc. [122,123]. To overcome these shortcomings, they investigated the various parameters that affect the growth of microgreens and the possibility of optimizing them for better results. The study concluded that microgreens could be grown for astronauts in special conditions and meet the nutritional requirements [121,124]. Further, they can also be used in improving cognitive function, which co-occurs with metabolic diseases, especially in the aged population [125], which can be addressed with microgreens as a source of good micronutrients.

## 8. Conclusions and Future Perspectives

Microgreens, a new plant-based functional food that consists of the seedlings of the edible plants harvested after 7–14 days of the germination process, are the stellar source of phytochemicals, such as essential minerals, polyphenols, carotenoids, chlorophyll, anthocyanins, glucosinolates, etc., which imparts high antioxidant, anti-inflammatory, anti-diabetic effects due to which it is considered as a practical food that might improve or attenuate chronic diseases. Diversity in the microgreen species offers a wide range of health benefits to consumers. However, there are certain challenges to overcome regarding the production, storage, and consumption of microgreens, such as maximizing their growth rate and yield potential. The microbial colonization and the diseases they cause need to be studied further, and a proper technique should be offered to avoid microbial invasion and, thus, prevent the possible occurrence of food-borne diseases. The idea of the metabolic profile of the different varieties of microgreens should be explored much further to map its activity. Additionally, there is increased attention towards microgreens production technology, the pre- and post-harvest techniques, packaging, and maintenance of shelf-life are yet to be discovered and optimized. On the other hand, the use of microgreens for the prevention and treatment of various chronic metabolic disorders discussed in this review is based on a limited number of studies. Additional investigation is warranted to determine the use of microgreens as a whole for their beneficial health-promoting properties with proof of mechanisms of action. Evidence-based research is required, and the use of microgreens as personal medicine is yet to be explored. In general, proper awareness about the nutritional characteristics, methods of preparation, sensory characteristics, and palatability of the microgreen communities should be provided to enhance health by reducing prevailing disorders. Further, the attributed health benefits of microgreens postulated are mainly based on their bioactive compounds [16,75,100]. The number of scientific studies that have measured the direct effect of health benefits from microgreens is minuscule, which is a major limitation, and it is important to conduct further studies to understand the microgreens’ individual and synergistic health effects.

## Figures and Tables

**Figure 1 molecules-28-00867-f001:**
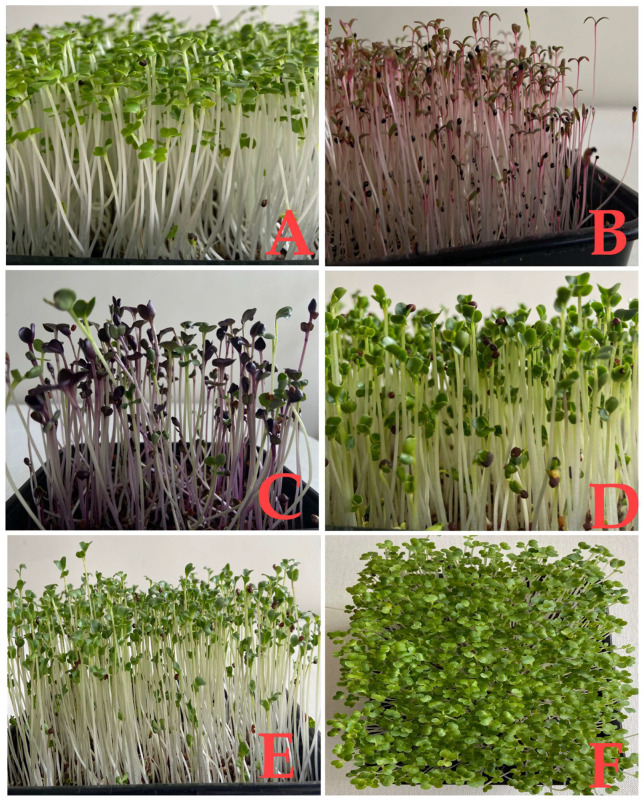
Different microgreens grown in a lab. (**A**) Bok choy, (**B**) Red amaranth, (**C**) Purple radish, (**D**) Kale, (**E**) Broccoli, and (**F**) Bok choy—top view.

**Figure 2 molecules-28-00867-f002:**
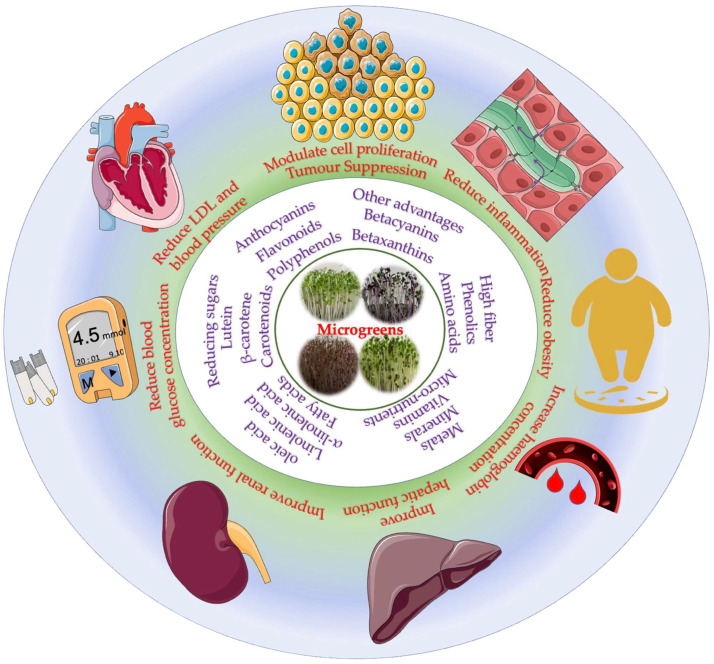
Overview of health benefits of microgreens.

**Table 1 molecules-28-00867-t001:** Comparison of sprouts, microgreens, baby greens, and mature plants [1,20,21,22].

Conditions	Sprouts	Microgreens	Baby Greens	Mature Plants
Height	5–8 cm	3–10 cm	10–15 cm	Several cm
Production time	3–10 days	7–21 days	20–40 days	Several months
Cultivation system	Do not require soil or medium to grow. Grow solely in water or in moisture.	Can be grown in soil or entirely in medium.	May or may not be grown in soil fields. Require medium to grow.	Grown in soil fields. Require medium to grow.
Light requirements	No, do not require light source.	Yes, requires light source.	Yes, requires light source.	Yes, requires light source.
Root appearance	Very tiny root without root hairs.	Small roots with root hairs.	Roots with root hairs.	Mature root system.
Agrochemicals use	No use of chemicals required.	No use of chemicals required.	Use of chemicals required.	Use of chemicals required.
Moisture/water use	Can be grown in little amount of water or even in little moisture content.	Small amount of water is required.	Water is required in large amount.	Abundant of water is needed.
Land space	Very small space is required for large scale production also.	Very small space is required for large scale production also.	Require a large area for their growth.	Grown over acres of free and open-spaced lands
Plant growth level at harvest time	Partial development of cotyledons with just the germinated seeds.	Full development of cotyledons with one or two true leaves.	Full development of young plant with true leaves.	Full development of mature plant that may bear fruits or vegetables.
Harvest type	No harvesting. Wholly edible.	Harvesting is done by removing the roots.	Removing the roots by cutting.	Harvesting is done by cutting the roots either manually or mechanically.

**Table 2 molecules-28-00867-t002:** Various nutrients and phytochemicals reported in different varieties of microgreen species.

Variety	Duration of Growth	Nutrients Reported	Health Benefits	Reference
Amaranth (Amaranthaceae)	10 days	chlorophyll a—0.25 mg/g chlorophyll b—0.20 mg/g carotenoids—0.023 mg/g anthocyanins—9 mg/100 g ascorbic acid—0.031 mg/g	antioxidant activity	[27,28]
Red Beets (Amaranthaceae)	10 days	polyphenols—313.8 mg/100 g betaxanthins—432.7 mg/100 g betacyanins—226.7 mg/100 g	antioxidant activity, gastrointestinal activity	[27]
Quinoa (Amaranthaceae)	–	tocopherols—65 μg/g tocotrienols—- β-carotene—738 μg/g fatty acids - α-linolenic acid—35.1% - linolenic acid—11.36% - palmitic acid & stearic acid—trace amounts - oleic acid—5.11%	antioxidant activity	[29]
Spinach (Amaranthaceae)	20 days	chlorophylls—44 μg/g lutein—54.2 μg/g β-carotene—44 μg/g phenols—632.3 μg/g ascorbic acid—130.5 μg/g	antioxidant activity	[30]
Swiss Chard (Amaranthaceae)	17 days	chlorophylls—0.771 mg/g carotenoids—0.122 μg/g phenolics—164 μg/g anthocyanins—11.78 μg/g sucrose—0.091 mg/g reducing sugars—0.75 mg/g sugars—2.2 mg/g	–	[31]
Onion (Amaryllidaceae)	10–12 days	ascorbic acid—29.9 mg/g α-tocopherols—15.2 mg/g β-carotene—3.8 mg/g oxalic acid—23.4 mg/g	–	[32]
Parsley (Apiaceae)	19 days	polyphenols—0.5 mg/g α-tocopherols—577.2 μg/g ascorbic acid—13.39 mg/g β-carotene—46.68 μg/g lutein—106.62 μg/g	antioxidant activity	[33]
Carrot (Apiaceae)	7–14 days	chlorophylls—290 μg/g polyphenols—250 μg/g anthocyanins—10 μg/g α-tocopherols—19 μg/g carotenoids—110 μg/g	antioxidant activity	[34]
Coriander (Apiaceae)	3–4 days	chlorophyll—13.36 mg/kg FW lutein—98.6 mg/k DW β-carotene—325.1 mg/kg DW ascorbic acid—121.40 mg/kg DW polyphenols—15.25 mg/g DW	–	[35]
Wild Rocket (Brassicaceae)	17 days	chlorophyll—1.007 mg/g carotenoids—0.171 μg/g phenolics—328 μg/g anthocyanins—8.83 μg/g sucrose—0.14 mg/g reducing sugars—1.4 mg/g sugars—4.4 mg/g	–	[31]
Radish (Brassicaceae)	9 days	ascorbic acid—52.31 mg/100 g phenolics –135.74 mg/100 g flavonoids—39.83 mg/100 g	–	[36]
Soybean (Fabaceae)	8 days	phenolics—5.5 mg/g flavonoids—68 mg/g	antioxidant activity	[37]
Cucumber (Cucurbitaceae)	9 days	ascorbic acid—24.01 mg/100 Phenolics—38.66 mg/100 g Flavonoids—17.15 mg/100 g	antioxidant activity	[36]
Jute (Malvaceae)	9 days	ascorbic acid—34.90 mg/100 g phenolics—152.10 mg/100 g flavonoids—142.39 mg/100 g	antioxidant activity	[36]
Leek (Amaryllidaceae)	–	sugars—0.5 mg/g ascorbic acid—9.1 mg/g polyphenols—31.5 mg/100 g chlorophylls—26.5 μg/g carotenoids—341.01 μg/g amino acids—755.4 mg/100 g	antioxidant activity, anti-obesity activity, anti-diabetic activity, and anti-cholinergic activity	[18]
Green peas (Leguminaceae)	–	sugars—0.5 mg/g ascorbic acid—9.1 mg/g polyphenols—108.5 mg/100 g chlorophylls—522.75 μg/g carotenoids—2794.4 μg/g amino acids—397.9 mg/100 g	antioxidant activity, anti-obesity activity, anti-diabetic activity, and anti-cholinergic activity	[18]

**Table 3 molecules-28-00867-t003:** Important conditions/requirements considered for growing microgreens.

Variables	Conditions/Requirements	Reference
Seeds	Seed quality and seeding density per tray play an important role to attain quality microgreens. Seed treatment effects germination percentage and weight of shoot	[56]
Light	Depending on species and variety of microgreens, light intensity has various impacts on growth. The red-to-blue light (445 nm) and far-red light (730 nm) have been demonstrated to raise carotenoid and photosynthetic pigments. 440 μ mol/m^2^/s an optimum photosynthetic photon flux (PPF) for the greater yield of microgreens. Fluctuation in PPF value (high or low) than optimal value results in reduced antioxidant and biomass concentrations. The normal development and the nutrient content of the microgreens are hindered when PPF value is low (110 μ mol/m^2^/s). While high PPF values (545 μ mol/m^2^/s) caused slight photo stress (photooxidation).	[37,57]
Growing medium	Mainly grown in a soilless substrate-based system, such as vermiculite, perlite, peat, and coconut coir dust, which shows enhanced growth of microgreens. Nutrient solution is supplied that contains all the essential ingredients for growth. Sometimes grown in water along with essential nutrients.	[6]
Pathogen treatment	A variety of plant infections can be a result of microorganisms due to the environmental conditions in which microgreens are grown, which may cause the roots and seedlings to rot. Trichoderma species are used for pathogen control in microgreens and their application as a seeding treatment also enhances the growth of microgreens. Calcium nitrate fertilizer application combined with a liquid fertilizer and nitrogen fertilizer treatment enhances growth of microgreens by 20%.	[58,59]
Harvesting	Microgreens can be harvested by trimming the plantlets either physically, using scissors or knife, or using automatic cutting tools. While harvesting, touching the growing medium should be avoided to reduce the contamination. Particles surrounding the seedlings are suggested to be removed as they adhered to cotyledons in many species.	[6]
Post-harvesting	Microgreens are cleaned, chilled to 5 °C for packaging using polythene bags to avoid contaminations. Handled according to all recommended good manufacturing practices to maintain hygiene and good quality. Packing in sterile rooms and maintaining sterile conditions increase the shelf-life of microgreens.	[60]

## Data Availability

Not applicable.

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
