# Peer review of "Microgreens—A Comprehensive Review of Bioactive Molecules and Health Benefits"

_molecules, 2023, doi:10.3390/molecules28020867_

Round 1

Reviewer 1 Report

Manuscript Number: Maharshi Bhaswant et al. -Molecules-2152777 

Manuscript Title:   Microgreens – A comprehensive review of bioactive molecules and health benefits 

General: In this manuscript authors review the applications of the microgreen, termed used to refer to food product developed from various commercial food crops like vegetables, grains, and herbs, consisting of developed cotyledons along with partially expanded true leaves. They focus on the use of microgreen to prevent diseases of the current generation due to the sedentary lifestyles and to develop health-promoting diets and habits with microgreens.

In general, the manuscript is interesting, timely, well written and it reads fluently. As a suggestion, along the text, authors write some long sentences which could be split into two and it would facilitate their understanding. 

On the other hand, the manuscript would benefit of a couple of figures summarizing i) the different varieties of microgreens, and ii) the effect of microgreens in metabolic health applications, and a table with the growth conditions and characteristics for the production of microgreens.

Author Response

  • As a suggestion, along the text, authors write some long sentences which could be split into two and it would facilitate their understanding. 

We thank reviewer for highlighting long sentences, we have now rephrased most of the long sentences throughout the manuscript. We have highlighted the changes using track changes.

  • On the other hand, the manuscript would benefit of a couple of figures summarizing i) the different varieties of microgreens, and ii) the effect of microgreens in metabolic health applications, and a table with the growth conditions and characteristics for the production of microgreens.

As suggested, we have now included Figure 1 (Different microgreens grown in lab) in section 4 (Different varieties of microgreens), Figure 2 (Overview of health benefits of microgreens) in section 7 (Effect of microgreens in metabolic health promoting applications) and Table 3 (Important conditions/requirements considered for growing microgreens) in section 6 (Growth conditions for the production of microgreens).

Reviewer 2 Report

Generally, the review article discusses an important issue, which the authors tacked and discussed adequately.

-        I suggest adding some images for a popular crop used for all purposes mentioned and compared by the authors: Sprouts vs. microgreens vs. baby greens vs. mature plants, to justify the differences in veg—growth characteristics.

-        If it is allowed by the journal, a list of contents could be added after the keywords to give a general idea about the whole review article.

-        I suggest adding a short hint about the methodology applied to collect and deal with the literature, mentioning the keywords used and the search engines, etc.

-        Check with the journal if the in-text citation is allowed, like “Di Bella et al. 2020” and “Xiao et al. 2015”

-        I tried to revise some English grammar and style errors till page 7; however, I suggest going for language editing to cover the whole manuscript

Line

Comment

37-39

Correct to be “Microgreens, known as “vegetable confetti”, are developed from the various commercial food crops like vegetables, grains, and herbs that basically consists of the fully developed cotyledons along with or without the partially expanded true leaves”

40

“and the probable presence present of the true leaves”

45

“to nature of plants”

55

Nutrients-enriched

56-57

where it they does not require a much of land space for its their cultivation

62

Delete “using

73

reliability in of the chemical

74

pose posing a serious threat among to the entire food chain

75

This The observed

81

, which are likely ….

92-93

Change to more fluent style such as “Consequently, they ate fresh fruits and vegetables and other kinds of green leafy vegetables as part of their daily diet to ensure adequate nutrition

94

has been led to

94

an abundant abundance of nutrients

Table 1

-        “Centimeters” instead of “inches”

-        “grow solely in water” instead of “Grows solely in water”

-        “Light requirements” instead of “Photosynthetic activity”

-        “roots pattern” or another proper expression instead of “Physiology of roots”

Table 2

-        I suggest removing “total” and “content” from the table and keep only names of the groups, just to have one format for the whole table

-        Some group names need to be in plural form (phenols or phenolics, anthocyanins, chlorophylls, amino acids,

216

I don’t understand the reason to combine Phenolic antioxidants and sugar content in one heading

178

“Suggesting that” instead of “suggested that”

182-183

“has been positively increased “ instead of “positively has increased”

192

“within the range of “ instead of “under the range of”

194

α-tocopherol is an extremely important

196

boosts boosting immune system, limiting free radical formation

198

vitamin E content in them thus

199

in the microgreens a group of

200

that belongs to

203

ranging between (11-76 μg/gm of microgreen).

207-208

Inducing induction of apoptosis

217

secondary antioxidants; I don’t know how do you classify them as secondary, where they are reported as primary antioxidants. I think you mean secondary metabolites

218

Preventing,  ……and reducing

221

and others associated

222

It is was also

223

ranged between

224

of microgreen especially, ..  which is was 10 times

225

Somebody (222222) suggested that; please refer to the authors here

229

Their studies suggested

230

free sugar content of was

231

and that is, while the lowest content was

Author Response

  • I suggest adding some images for a popular crop used for all purposes mentioned and compared by the authors: Sprouts vs. microgreens vs. baby greens vs. mature plants, to justify the differences in veg—growth characteristics.

As this review is mainly focused on microgreens rather than sprouts, baby greens and mature plants, we have not included the image as suggested by the reviewer. However, we have included Figure 1 (Different microgreens grown in lab) in section 4 (Different varieties of microgreens).

  • If it is allowed by the journal, a list of contents could be added after the keywords to give a general idea about the whole review article.

At the end of introduction (line 87-90), we have provided a brief overview on sections included in review to get the general idea for readers.  

  • I suggest adding a short hint about the methodology applied to collect and deal with the literature, mentioning the keywords used and the search engines, etc.

We thanks the reviewer for his suggestion on adding methodology for this review. We have now included a small section “Literature search” (line 91-96) on methodology.  

  • Check with the journal if the in-text citation is allowed, like “Di Bella et al. 2020” and “Xiao et al. 2015”

We have referred the instructions to authors page of the journal “Molecules”, there is no mention of in-text citation is not allowed. We have also referred two recent reviews published in January, 2023 in this journal and confirmed it is allowed.

  • I tried to revise some English grammar and style errors till page 7; however, I suggest going for language editing to cover the whole manuscript.

We thank reviewer for highlighting our grammatical, typographical, and overlooked errors. Apart from the suggested list, we have revised all sections of the manuscript in order to improve language and reduce the grammatical and style errors. We have highlighted the changes using track changes.